# Stabilization of Different Soil Types Using a Hydraulic Binder

Fawzia Kired, Miloš Šešlija [ID], Tiana Milović, Anka Starčev-Ćurčin *[ID], Vesna Bulatović *[ID] and Nebojša Radović

Faculty of Technical Sciences, University of Novi Sad, 21000 Novi Sad, Serbia; fawziamuftah74@gmail.com (F.K.);
sele@uns.ac.rs (M.Š.); tiana.milovic@uns.ac.rs (T.M.); radovicn@uns.ac.rs (N.R.)
* Correspondence: astarcev@uns.ac.rs (A.S.-Ć.); vesnam@uns.ac.rs (V.B.)

**Abstract:** This paper presents an analysis of the stabilization of different soil types using a hydraulic binder. A study was carried out on soils that can be classified into two groups: cohesive and non-cohesive soils. Clay soils of medium and low plasticity according to the USCS classification were used as cohesive materials, while the sandy material containing dust was considered as non-cohesive material. Samples were taken from fifteen locations in Vojvodina province, Serbia. A hydraulic binder was used as a binder based on cement and lime. The amounts of the binder were estimated at 3, 5, 7, and 9%. In order to determine the basic physical and mechanical characteristics of the specimens, the following tests were performed: unconfined compressive strength after 7 and 28 days, indirect tensile strength after 7 and 28 days, as well as the California Bearing Ratio. Based on the obtained results, it can be concluded that increasing the amount of binder results in an increase in the subgrade load-bearing capacity. However, it should be emphasized that the subgrade containing non-cohesive material had a lower growth in the load-bearing capacity than those with the cohesive material.

**Keywords:** soil stabilization; hydraulic binder; unconfined compressive strength; indirect tensile strength; California Bearing Ratio; Proctor test





## 1. Introduction

The strength of the pavement subgrade (i.e., natural/existing soil layer under the pavement structure) plays an important role in achieving a quality and economical pavement design. In the case of weak subgrade with poor engineering properties, soil stabilization represents the solution that provides a reduction in pavement thickness and improves subgrade bearing capacity compared to natural soil [1]. According to IRC:SP:89 2010 [2], soil stabilization is the process of blending and mixing materials with a soil in order to improve certain of its properties. The above-mentioned can be achieved by applying mechanical and chemical methods. Wei et al. [3] stated that there is also a third, biological method, such as microbially induced calcium carbonate precipitation, which was extensively studied, but its effectiveness in practical applications remained uncertain [4,5]. Mechanical stabilization implies blending soils of two or more gradations or mixing soil with aggregates to obtain a material that satisfies the required specification, while chemical (i.e., additive) stabilization is achieved by the addition of proper percentages of binder (i.e., cement, lime, fly ash, etc.) or combinations of those binders to the soil after which the spreading, sprinkling of water, and compaction at optimum moisture content are achieved by conventional means [2]. The primary ingredient necessary for stabilizing soil and improving its engineering properties is calcium [6]. Both cement and lime still lead in geotechnical practice as traditional binders for improving the engineering properties of weak soils [7] due to calcium providing low cost and significant improvement of the soil strength [3]. Although lime may have some advantage in reducing the plasticity index ($I_P$) of highly plastic soils, Portland cement improves the strength by forming C-S-H, generates Ca $(OH)_2$, and reduces $I_P$ [6]. However, it has been reported that sulfate-rich soils treated with either ordinary Portland cement or lime have the potential to expand, i.e., lime or ordinary Portland cement treatment of a soil containing sulfates represents a potential risk to its stability and durability [8–11]. For that

reason, the usage of alternative hydraulic binders is recommended [8]. Generally, when locally available industrial by-products, agricultural wastes, or natural materials are used, stabilized soils represent highly useful, cost-effective, and sustainable construction materials [12]. Industrial and agricultural by-products and wastes such as fly ash, blast-furnace slag, silica fume, red mud, cement kiln dust, plastic waste, calcium carbide residue, rice husk ash, sugarcane bagasse ash, etc., and natural materials such as zeolite, bentonite clay, etc., are found to be useful in pavement construction and soil stabilization as alternatives for traditional stabilizers, i.e., ordinary Portland cement and lime.

In recent years, a large number of authors have focused their research on soil stabilization. Tanzadeh et al. [13] investigated the effect of lime powder, added in 0, 2, 4, 8, and 16% of the dry soil weight, on the behavior of kaolinite clay soil. An optimum lime percentage (4%) was defined by determining the plastic limit. Specimens containing an optimum rate of lime powder were examined using unconfined compressive strength (UCS) tests and California Bearing Ratio (CBR) over a curing period of 90 days. Young's modulus was determined based on UCS testing and the definition of the stress–strain curve. The results showed a significant positive impact of lime in increasing the maximum UCS value as well as increasing the CBR value and Young's modulus of stabilized soil. Baldovino et al. [14] determined the ratio between the splitting tensile strength ($q_t$) and the UCS ($q_u$) of clayey soil in the metropolitan region of Curitiba City, Brazil, which was treated with different lime contents and curing times. It was observed that the $q_t/q_u$ ratio was between 0.17 and 0.20 in relation to the curing time, and an exponential relation existed between them. Meanwhile, the UCS of lime-treated soil was found to be approximately four times the initial value. Ghobadi et al. [15] conducted a geotechnical investigation on lime-treated clay soils from Hamedan City, Iran. Lime was added in different percentages (1, 3, 5, and 7%) and UCS values were determined after curing times of 7, 15, 30, and 45 days. The results indicated that with an increase in the content of lime up to 7%, there was an increase in UCS. The optimum lime content and proper curing time for lime-treated clay soils were at least 7% and 30 days, respectively. Negawo et al. [16] studied highly expansive clay soils from the Highlands of Ethiopia to evaluate the efficiency of lime treatment in order to improve their mechanical properties for road subgrades. Soils treated with quicklime at 5, 7, and 9%, by dry weight of the soil, were cured for 7 days under a controlled temperature of $(40 \pm 2)$ °C and geomechanical laboratory tests were conducted to evaluate its impact on the engineering properties of the soil (e.g., UCS, CBR). Based on the obtained results, it was concluded that expansive soils of the studied area can be effectively stabilized for road subgrade works with the addition of 7% quicklime by dry weight of the soil. Okonkwo and Kennedy [17] investigated the effects of cement and lime on the mechanical properties of subgrade which consisted of black cotton soil. Black cotton soil, i.e., expansive soil, is a type of soil that has a high clay content and tends to have significant volume changes due to changes in moisture content. Based on the obtained results, it was concluded that both cement and lime were effective stabilizing agents that increased the optimal moisture content. The engineering properties of the stabilized soil subgrade were also evaluated, and the use of cement and lime as stabilizers was found to be effective in improving soil characteristics for subgrade applications, due to increased maximum dry density values, reduced plasticity index values, and increased CBR and UCS values. Khemissa and Mahamedi [18] conducted a series of normal Proctor compaction tests, CBR tests, and undrained direct shear tests on Sidi-Hadjrès (Algeria) expansive overconsolidated clay treated with a mixture of various cement and lime contents and compacted under the optimum Proctor conditions. The results showed that the geotechnical parameter values confirmed the bearing capacity improvement of this natural clay, which was characterized by a significant increase in soil strength and durability, and the best performances were obtained for a mixed treatment corresponding to 8% cement and 4% lime contents. Lebo et al. [6] investigated the stabilization of Zagreb clayey soil with different types of binders (cement, slag, and fly ash). Under laboratory conditions, composite samples were prepared, where the binders were added to the clay in ratios of 5, 10, and 20% and tested at different curing time intervals of 7, 14,

and 28 days. The influence of different types and amounts of binder and the age of the composite sample on UCS was examined and analyzed. The obtained results showed that the utilization of cement, fly ash, and slag can improve the mechanical properties of the used clay, depending on the amount of binder and the curing time of the composite. Rai et al. [19] evaluated the effectiveness of the usage of cement and fly ash in the stabilization of subgrade soils. In this investigation, soil subgrade samples were taken from the Toll Plaza National Highway in Hyderabad, Pakistan, and had just 5.28% moisture content. The soil was grey in color and it was classified as A-2-4 (in this case, 70.49% silt, 13.76% clay, and 15.78% sand). The following mix proportions were examined: natural soil, 5% fly ash and 2% cement, 10% fly ash and 4% cement, 15% fly ash and 6% cement, 20% fly ash and 8% cement. The CBR value of the initial subgrade soil was 2.9, while with the inclusion of fly ash and cement, values were further improved. The optimum percentage of mixture (20% fly ash and 8% cement) gave a CBR value of 10.12, the highest result for soil subgrade. Maximum UCS values were obtained for the mixture consisted of 20% fly ash and 8% cement, after 1, 7, and 14 days of treated soil curing time. Caselles et al. [8] studied the stabilization of soils containing sulfates by adding alternative hydraulic binders such as ground granulated blast furnace slag, which possesses latent hydraulic properties [20]. Caselles et al. [8] observed that treatment of sulfate-rich soil with cementitious binders having high $C_3A$ content led to volume expansions greater than 5%, while treatments with binders containing a high fraction of ground granulated blast furnace slag showed volume expansions of less than 5% and about 89% of sulfates were immobilized in the solid matrices. These preliminary results confirmed that slag binders are effective for the treatment of soils containing sulfates. Ahmad et al. [21] carried out the chemical stabilization of Malaysian peat soil with ordinary Portland cement, with different percentages of silica fume and with different percentages of ordinary Portland cement and silica fume mix. UCS and CBR tests were carried out after 7, 14, and 28 days of curing. The incorporation of silica fume and ordinary Portland cement brought a substantial improvement in the mechanical properties of the stabilized peat soil. The highest UCS value was obtained at 20% silica fume, and an unsoaked CBR value of 42.95 was observed using 15% silica fume and 15% ordinary Portland cement after 28 days of curing.

Barišić et al. [22] investigated the properties of three biomass ashes used as additives to lime-stabilized low-bearing soil (low-plasticity clay) for embankment and subgrade purposes. The results indicated that there is potential for using barley, sunflower seed shells, and wheat fly ash as lime substitutes in the soil stabilization of road works.

In Western Europe, particularly in countries like France, Belgium, and Germany, the usage of hydraulic road binders (HRBs) is widely prevalent and surpasses that of cement for soil stabilization. HRBs are factory-produced blends of two or more component materials (e.g., lime, cement, ground granulated blast furnace slag, coal fly ash, gypsum, etc.) [23], used as stabilizers for treatment of road base, subbase, as well as subgrade. T Wang and Baaj [24] investigate the chemical and physical properties of organic clayey subgrade materials treated by different HRBs. Test results indicated that all the three subgrade soils were fine-grained soils with substantial silt and clay particles and organic matter. Cement and HRBs significantly improved soil engineering properties, while HRB-treated subgrade soils had lower sulfate contents than cement-treated soil. Among all HRB-treated materials, with abundant curing, the subgrade soils treated with the HRB composed of Portland limestone cements and ground granulated blast furnace slag had the highest UCS values, followed by the soil treated with HRB composed of Portland limestone cements and fly ash and HRB composed of cement for general use recommended by Canadian Standard CAN/CSAA3001-13 [25] and ground granulated blast furnace slag.

In the case of the natural materials, zeolites are highly used as a partial replacement for cement or lime in the stabilization of different soil types. Kushawa and Yadav [26] examined the effect of clinoptilolite (type of natural zeolite) on the mechanical properties of cemented sand taken from the Narmada River in India. The results showed that adding clinoptilolite at the optimum value of 30% to cemented sand UCS increased stability for all the mixtures

with different cement contents (4, 6, 8, and 10%). Chenarboni et al. [27] investigated the effect of partial cement replacement with zeolite on the mechanical properties of expansive clay. The maximum UCS was obtained for the mixture that contained 12% cement with 30% zeolite as cement replacement, after 28 days of curing. On the other hand, Shirmohammadi et al. [28] studied the effect of lime–zeolite stabilization on the behavior of a silt-sized natural soil of low plasticity, by performing freezing and thawing cycles and determination of UCS of the composite samples. The results indicated that the partial replacement of lime with zeolite caused an increase in the strength of the mixtures up to 30% (which was considered as the optimum zeolite replacement), beyond which it decreased. The lime–zeolite composite material showed better durability to the freezing–thawing process compared to the untreated natural soil. Akbari et al. [29] stabilized samples of soft soil (Zenoz kaolinite of Tabriz in northwestern Iran) with 5, 10, and 15% modifier lime, lime–nano-zeolite, and lime–nano-zeolite–polypropylene fiber, and subjected them to 1–7 wet–dry cycles. The results indicated that the optimal replacement of lime with nano-zeolite was 40%, and the optimal amount of polypropylene fiber inclusion was 1% in the stabilized soil matrix. Furthermore, the results showed that the specimen containing 15% lime–nano-zeolite–polypropylene fiber had excellent durability against environmental conditions and very good performance in terms of UCS, tensile strength, and weight loss. Besides natural zeolites, bentonite clay can also be used as a soil stabilizer. Baker et al. [30] used a bentonite clay–water slurry due to its cohesive and eco-friendly nature to improve sand strength by means of manual injection in the laboratory and pilot scales. Sand was stabilized using 0%, 1%, 2%, 3%, and 4% bentonite clay (by weight of dry sand), at different curing times (0, 1, 2, and 3 days). The test results showed that the slurry composed of 3% bentonite clay additive with 10.3% added water by weight of dry sand mass were the optimum amounts for the stabilization process, which provided a substantial resistance to shear forces.

From the review presented here, lime and cement soil stabilization works by binding, and practically all soil types are compatible with this type of soil stabilization. Despite the results presented in existing studies on blended binders and their applications in various regions with diverse soil types, it is still unclear how these methods behave when used in specific conditions. Furthermore, there is a lack of knowledge regarding the use of mixed combinations of binders for the stabilization of weak soils, which experimentally include novel types of binders, such as fly ash, bio fly ash, or slag. Geographical regions often dictate the use of binders for soil stabilization. In most regions, lime and cement are readily available and thus more attractive for application. For these reasons, the utilization of a mixed binder based on cement and lime is still easier and more practical to use. However, we should increasingly move towards the use of alternative binders that are more environmentally friendly.

The aim of the experimental research presented in this paper was to evaluate a hydraulic binder based on cement and lime (BeoBond C30, LaFarge Beočin, Serbia) for the stabilization of different soil types. Soil samples were taken from fifteen locations in Vojvodina province, Serbia. The amounts of the binder were estimated at 3, 5, 7, and 9%. In order to determine the basic physical and mechanical characteristics of the samples, the following tests were performed: UCS after 7 and 28 days of curing, indirect tensile strength (ITS) after 7 and 28 days of curing, as well as the CBR.

## 2. Materials and Methods

### 2.1. Component Materials

The materials used in this study include soil subgrade samples collected from the places where the construction of future traffic roads is planned, and their positions are marked in Figure 1. According to the geological map of Vojvodina province [31], the soils of the Pannonian Plain were formed mainly on sedimentary rocks, except for the soils on Fruška Gora and Vršački breg, which have mostly metamorphic and partly igneous rocks as their geological base. The relief of Vojvodina was formed by the action of endogenous

and exogenous forces. By the action of endogenous forces, the morphostructural forms of the relief were formed, i.e., the basic contours of the present-day relief were created. However, the morphostructural forms of the relief are largely exogenously reshaped.

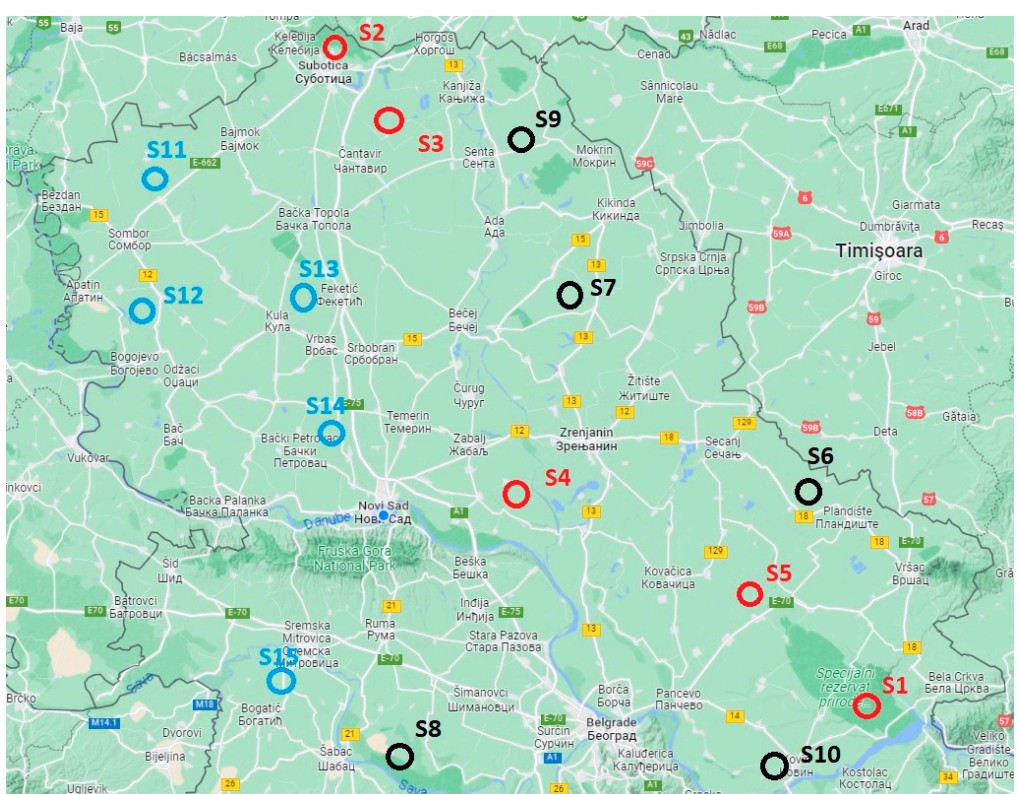

**Figure 1.** Places in the province of Vojvodina from which soil samples were taken.

After the sampling of the subgrade soils, their granulometric compositions were determined in order to classify the soils in terms of the granulation, the consistency ($W_L$—Liquid limit, $W_P$—Plastic limit and $I_P$—Plasticity index), and the type according to American Association of State Highway and Transportation Officials (AASHTO) [32] and Unified Soil Classification System (USCS) [33]. The granulometric compositions of the soils were determined in accordance with the standard SRPS EN 17892-4 [34], and the obtained results are shown in Table 1.

**Table 1.** Granulometric soil composition, soil consistency, and AASHTO and USCS classifications.

| Samples | | Granulometric Composition (%) | | | | Consistency (%) | | | AASHTO | USCS |
|---|---|---|---|---|---|---|---|---|---|---|
| | | Clay | Dust | Sand | Gravel | $W_L$ | $W_P$ | $I_P$ | | |
| I batch | S1 | 11.59 | 82.40 | 6.01 | - | 45.90 | 24.90 | 21.00 | A-7-6 | CL |
| | S2 | 12.80 | 82.20 | 5.00 | - | 45.90 | 24.90 | 21.00 | A-7-6 | CL |
| | S3 | 16.70 | 81.30 | 2.00 | - | 55.70 | 31.00 | 24.70 | A-7-5 | MH |
| | S4 | 12.65 | 85.00 | 2.35 | - | 53.95 | 30.90 | 23.05 | A-7-5 | MH |
| | S5 | 10.90 | 86.00 | 3.10 | - | 54.85 | 32.90 | 21.95 | A-7-5 | MH |
| II batch | S6 | 1.90 | 62.10 | 29.30 | 6.70 | - | - | - | A-4 | ML |
| | S7 | 5.00 | 65.00 | 27.70 | 2.30 | - | - | - | A-4 | ML |
| | S8 | 5.50 | 72.00 | 22.50 | - | 29.05 | 20.90 | 8.15 | A-4 | CL |
| | S9 | 7.90 | 73.90 | 18.20 | - | 34.00 | 24.90 | 9.10 | A-4 | CL |
| | S10 | 7.90 | 76.80 | 15.30 | - | 36.90 | 27.00 | 9.90 | A-4 | CL |

**Table 1.** *Cont.*

| Samples | | Granulometric Composition (%) | | | | Consistency (%) | | | AASHTO | USCS |
|---|---|---|---|---|---|---|---|---|---|---|
| | | Clay | Dust | Sand | Gravel | $W_L$ | $W_P$ | $I_P$ | | |
| III batch | S11 | 8.00 | 70.10 | 21.90 | - | 41.80 | 30.90 | 10.90 | A-2-7 | CL |
| | S12 | 8.90 | 73.80 | 17.30 | - | 43.00 | 30.90 | 12.10 | A-7-5 | CL |
| | S13 | 5.10 | 82.90 | 12.00 | - | 40.05 | 29.00 | 11.05 | A-6 | CL |
| | S14 | 5.10 | 78.10 | 16.80 | - | 43.90 | 29.00 | 14.90 | A-7-6 | CL |
| | S15 | 5.90 | 77.90 | 16.20 | - | 44.00 | 30.00 | 14.00 | A-7-6 | CL |

$W_L$—Liquid limit, $W_P$—Plastic limit, $I_P$—Plasticity index, CL—Low-plasticity clays with a liquid limit of less than 50%, ML—Dusts of low plasticity with a liquid limit of less than 50%, MH—high-plasticity dust with a liquid limit over 50%, A-2-7—Includes granular materials containing 35% of the fraction that passes through a 0.075 mm sieve and part of the fraction that passes through a 0.425 mm sieve. Where the plasticity index is greater than 10%, A-4 represents non-plastic or medium-plasticity dusty materials, which typically have 75% or more fractions passing through a 0.075 mm sieve, A-6 represents plastic clay materials, which usually have 75% or more fractions passing through a 0.075 mm sieve, A-7-5 includes those materials of medium plasticity index in relation to their yield strength, which can be very elastic and subject to a significant change in volume, A-7-6 includes those materials that have high plasticity indices in relation to their liquid limit, which are subject to extremely large volume changes.

According to the obtained results shown in Table 1, samples from S1 to S5 (i.e., the first batch of samples) belong to the same group of materials, i.e., dusty clays, and an increased share of clay particles can be observed compared to the other samples. The samples marked from S6 to S10 (i.e., the second batch of samples) are classified as cohesive soils containing sandy components and much less clay particles, while materials from the third group of samples, from S11 to S15 (i.e., third batch of samples), belong to clays of low plasticity or the so-called loess material. In addition to the basic tests of the materials, additional tests were performed to determine optimal moisture content $W_{opt}$ (i.e., the amount of water at which the maximum dry density $\gamma_{d,max}$ is achieved), as well as the CBR. Obtained test results are shown in Table 2.

**Table 2.** $W_{opt}$, $\gamma_{d,max}$, and CRB of the soil samples.

| Samples | | $W_{opt}$ (%) | $\gamma_{d,max}$ (Mg/m$^3$) | CBR (%) |
|---|---|---|---|---|
| I batch | S1 | 18.90 | 1.605 | 3.2 |
| | S2 | 19.70 | 1.600 | 2.8 |
| | S3 | 22.20 | 1.495 | 2.3 |
| | S4 | 24.10 | 1.501 | 1.9 |
| | S5 | 22.00 | 1.499 | 1.5 |
| II batch | S6 | 15.80 | 1.803 | 18.1 |
| | S7 | 15.70 | 1.699 | 17 |
| | S8 | 17.90 | 1.707 | 14.9 |
| | S9 | 17.90 | 1.698 | 15.2 |
| | S10 | 19.90 | 1.702 | 14 |
| III batch | S11 | 19.20 | 1.597 | 7.1 |
| | S12 | 21.00 | 1.642 | 6.3 |
| | S13 | 20.10 | 1.592 | 5.1 |
| | S14 | 19.00 | 1.642 | 4.5 |
| | S15 | 20.10 | 1.599 | 4.1 |

Reviewing the results showed that the CBR value ranges from 1.5 to 3.2%, 14 to 18.1%, and 4.1 to 7.1% for the first, second, and third batches of samples, respectively. From the presented results, it can be seen that dusty clay and loess material had reduced CBR, while sandy dust samples had significant values of bearing capacity (over 10%).

The hydraulic binder BeoBond C30 (LaFarge, Serbia) based on cement and lime was used as the stabilizer. The following tests were conducted to assess the binder's properties:

- Standard consistency, (%) [35];
- Initial setting time, (min) [35];

- Compressive strength after 2 days, (MPa) [36];
- Compressive strength after 28 days, (MPa) [36];
- Soundness—Le Shatelier, (mm) [35];
- Fineness, mass fraction of the residue on the 90 μm sieve, (%) [35].

The test results as well as the prescribed requirements by the standard SRPS EN 13282-2 [37] are shown in Table 3.

**Table 3.** Physicomechanical properties of the hydraulic binder.

| Property | Requirement | Obtained Value |
|---|---|---|
| Standard consistency, (%) | - | 37.5 |
| Initial setting time, (min) | ≥150 | 180 |
| Compressive strength after 2 days, (MPa) | - | 15.0 |
| Compressive strength after 28 days, (MPa) | ≥32.5, ≤52.5 | 48.0 |
| Soundness—Le Shatelier, (mm) | ≤10 | 0.7 |
| Fineness, mass fraction of the residue on the 90 μm sieve mesh, (%) | ≤15 | 13.0 |

According to the results presented in Table 3, it can be seen that the tested binder met all the requirements prescribed by the standard SRPS EN 13282-2 [37].

### 2.2. Methods

Based on the results of testing the component materials, the production of trial soil–binder mixtures was planned. Five different hydraulic binder contents (0, 3, 5, 7, and 9%) were designated containing the soils from fifteen different locations, and a total of 75 different mixtures were prepared and examined. Figure 2 shows preparation of the soil mixture with the binder.

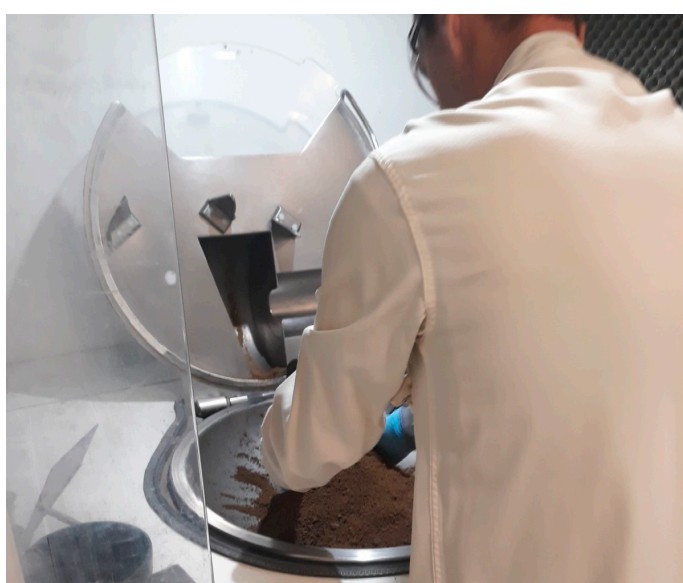

**Figure 2.** Preparation of the soil–binder mixture.

The following tests were conducted on specimens:

- Consistency testing;
- Proctor compaction test;
- UCS test;
- CBR test;
- ITS test.

### 2.2.1. Consistency

The Atterberg limits are commonly used in geotechnical engineering to classify and characterize fine-grained soils [33]. Therefore, on all mixtures, except those containing soils S6 and S7 (i.e., S6 and S7 were characterized as the non-cohesive soils according to the results shown in Table 1), the determination of the liquid limit and plastic limit was performed as well as the determination of the plasticity index. The tests were conducted in accordance with the standard SRPS U.B1.020 [38].

### 2.2.2. Proctor Compaction Test

Proctor compaction tests were carried out on each soil–binder mixture in accordance with standard SRPS EN 13286-2 [39] to obtain the $\gamma_{d,max}$ and $W_{opt}$ of the stabilized soils. During the preparation of the samples, a compaction energy of 600 kN/m$^3$ was used and the samples were prepared in three layers using a standard mold with a diameter of 102 mm and a height of 116 mm.

### 2.2.3. CBR Test

The CBR test represents a method for evaluating the stability of soil subgrade. The test was performed according to the procedure described in the standard SRPS EN 13286-47 [40]. Specimens stored in water for later testing are shown in Figure 3. A total of 75 specimens were prepared and cured for 4 days before the standard piston was used to penetrate the specimens up to 2.54 mm.

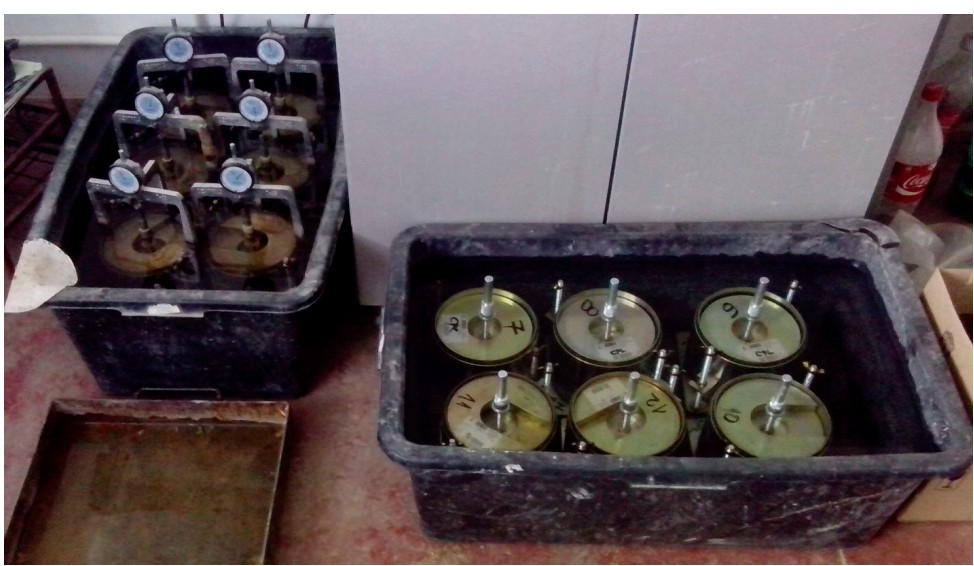

**Figure 3.** CBR test specimens.

### 2.2.4. UCS Test

The UCS values of various combinations of hydraulic binder and different soil types were determined in accordance to the standard SRPS EN 13286-41 [41]. For the purpose of this examination, specimens with a diameter of 100 mm and a height of 200 mm were made, which fulfilled the requirement that the height-to-diameter ratio be 2:1.

The specimens were prepared in a standard Proctor test mold and compacted in five layers. All prepared specimens were curried in a humid chamber for 7 and 28 days when their UCS values were determined. Figure 4 represents a UCS test specimen.

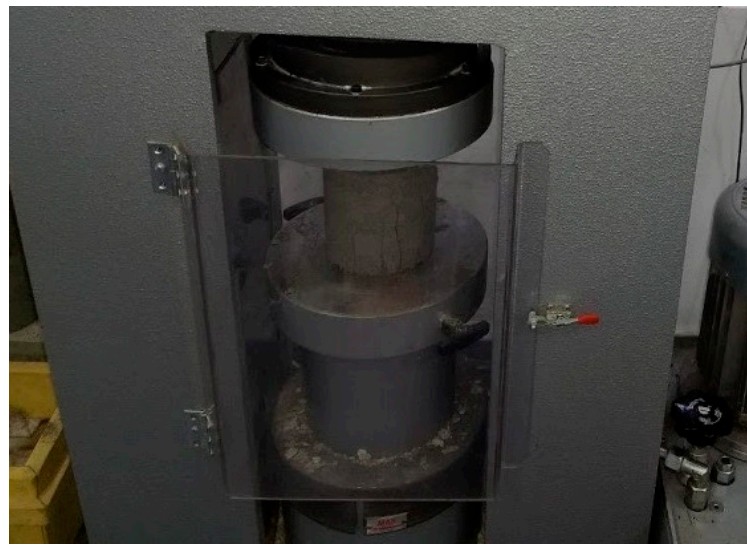

**Figure 4.** UCS testing of the specimen.

### 2.2.5. ITS Test

The ITS test was performed in accordance with the standard SRPS EN 13286-42 [42]. Cylindrical specimens with a diameter of 102 mm and a length of 116 mm, specially prepared in a standard mold according to the Proctor method in three layers, were used for this test. The specimens prepared in this way were kept in a humid chamber for 7 and 28 days before the testing. In Figure 5 is shown ITS testing on a specimen.

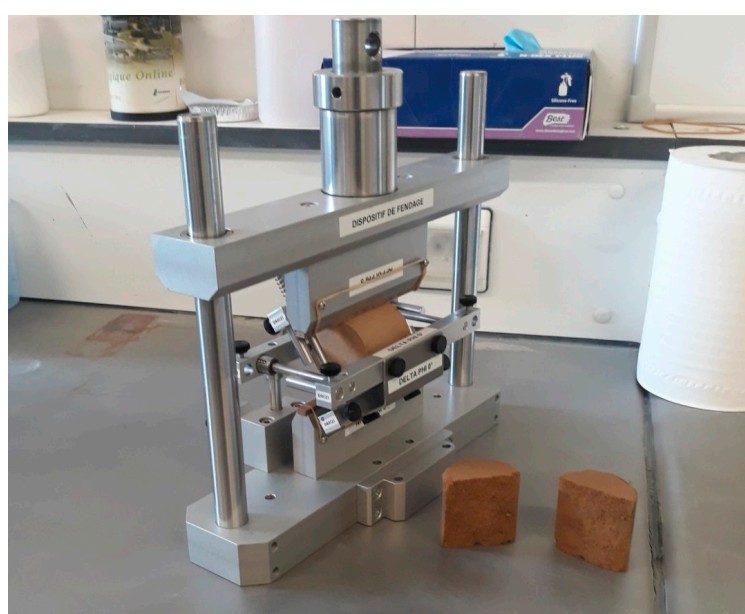

**Figure 5.** ITS testing of a specimen.

### 3. Results and Discussion

The statistical analysis of the obtained results was carried out for all tests of mixtures that were classified into three groups (i.e., batches) according to soil types, and it is presented in Table 4. The first batch of specimens contained soils from S1 to S5, the second batch consisted of soil specimens marked from S6 to S10, and the third batch consisted of S11 to S15. This means that each batch has a minimum of five specimens, except for UCS and ITS testing, where the values are obtained based on all the soil specimens, e.g., from S1 to S5, for which three specimens were made and tested after 7 days and three specimens

were made and tested after 28 days, which means a total of 15 specimens per each level of binder addition.

**Table 4.** Presentation of statistical processing of data from this research.

| Tested Property | | | Hydraulic Binder Content (%) | | | | |
|---|---|---|---|---|---|---|---|
| | | | 0 | 3 | 5 | 7 | 9 |
| I batch of specimens (with S1–S5) | $W_L$ | $\bar{x}$ (%) | 51.26 | 51.59 | 51.54 | 51.39 | 51.51 |
| | | $\sigma$ (%) | 4.93 | 4.99 | 4.98 | 4.93 | 4.91 |
| | | $C_v$ (%) | 9.62 | 9.68 | 9.67 | 9.60 | 9.53 |
| | $W_P$ | $\bar{x}$ (%) | 28.92 | 29.10 | 29.13 | 29.05 | 29.13 |
| | | $\sigma$ (%) | 3.76 | 3.81 | 3.80 | 3.78 | 3.78 |
| | | $C_v$ (%) | 12.99 | 13.09 | 13.05 | 13.00 | 12.98 |
| | $I_P$ | $\bar{x}$ (%) | 22.34 | 22.49 | 22.41 | 22.34 | 22.38 |
| | | $\sigma$ (%) | 1.57 | 1.52 | 1.59 | 1.53 | 1.52 |
| | | $C_v$ (%) | 7.01 | 6.76 | 7.09 | 6.84 | 6.77 |
| | $W_{opt}$ | $\bar{x}$ (%) | 21.38 | 22.18 | 21.64 | 21.86 | 21.96 |
| | | $\sigma$ (%) | 2.09 | 2.13 | 2.26 | 2.37 | 2.36 |
| | | $C_v$ (%) | 9.76 | 9.58 | 10.44 | 10.82 | 10.76 |
| | $\gamma_{d.max}$ | $\bar{x}$ (Mg/m$^3$) | 1.54 | 1.58 | 1.60 | 1.61 | 1.64 |
| | | $\sigma$ (Mg/m$^3$) | 0.06 | 0.04 | 0.04 | 0.04 | 0.03 |
| | | $C_v$ (%) | 3.71 | 2.74 | 2.53 | 2.23 | 1.89 |
| | CBR | $\bar{x}$ (%) | 2.34 | 16.78 | 57.76 | 112.40 | - |
| | | $\sigma$ (%) | 0.68 | 1.24 | 3.06 | 3.73 | - |
| | | $C_v$ (%) | 29.08 | 7.39 | 5.30 | 3.32 | - |
| | 7-day UCS | $\bar{x}$ (MPa) | 0.00 | 0.00 | 0.41 | 1.14 | 1.81 |
| | | $\sigma$ (MPa) | 0.00 | 0.00 | 0.00 | 0.02 | 0.02 |
| | | $C_v$ (%) | 0.00 | 0.00 | 1.18 | 2.10 | 1.27 |
| | 28-day UCS | $\bar{x}$ (MPa) | 0.00 | 0.67 | 0.92 | 1.42 | 2.90 |
| | | $\sigma$ (MPa) | 0.00 | 0.01 | 0.01 | 0.01 | 0.01 |
| | | $C_v$ (%) | 0.00 | 1.71 | 1.24 | 0.81 | 0.39 |
| | 7-day ITS | $\bar{x}$ (MPa) | 0.00 | 0.00 | 0.00 | 0.15 | 0.30 |
| | | $\sigma$ (MPa) | 0.00 | 0.00 | 0.00 | 0.01 | 0.01 |
| | | $C_v$ (%) | 0.00 | 0.00 | 0.00 | 7.81 | 3.75 |
| | 28-day ITS | $\bar{x}$ (MPa) | 0.00 | 0.00 | 0.13 | 0.21 | 0.42 |
| | | $\sigma$ (MPa) | 0.00 | 0.00 | 0.01 | 0.02 | 0.02 |
| | | $C_v$ (%) | 0.00 | 0.00 | 9.05 | 7.53 | 4.28 |
| II batch of specimens (with S6–S10) | $W_L$ | $\bar{x}$ (%) | 33.32 | 33.50 | 33.72 | 33.72 | 33.83 |
| | | $\sigma$ (%) | 3.97 | 3.74 | 3.52 | 3.44 | 3.46 |
| | | $C_v$ (%) | 11.91 | 11.15 | 10.44 | 10.21 | 10.24 |
| | $W_P$ | $\bar{x}$ (%) | 24.27 | 24.33 | 24.33 | 24.33 | 24.43 |
| | | $\sigma$ (%) | 3.10 | 3.15 | 3.15 | 3.10 | 3.06 |
| | | $C_v$ (%) | 12.77 | 12.96 | 12.96 | 12.76 | 12.50 |
| | $I_P$ | $\bar{x}$ (%) | 9.05 | 9.17 | 9.38 | 9.38 | 9.40 |
| | | $\sigma$ (%) | 0.88 | 0.60 | 0.45 | 0.45 | 0.52 |
| | | $C_v$ (%) | 9.68 | 6.58 | 4.78 | 4.78 | 5.53 |
| | $W_{opt}$ | $\bar{x}$ (%) | 17.44 | 17.30 | 17.59 | 17.67 | 17.76 |
| | | $\sigma$ (%) | 1.75 | 1.80 | 1.69 | 1.68 | 1.69 |
| | | $C_v$ (%) | 10.01 | 10.38 | 9.63 | 9.50 | 9.51 |
| | $\gamma_{d.max}$ | $\bar{x}$ (Mg/m$^3$) | 1.72 | 1.79 | 1.82 | 1.84 | 1.86 |
| | | $\sigma$ (Mg/m$^3$) | 0.05 | 0.05 | 0.05 | 0.05 | 0.05 |
| | | $C_v$ (%) | 2.64 | 2.79 | 2.64 | 2.81 | 2.93 |
| | CBR | $\bar{x}$ (%) | 15.84 | 25.16 | 41.34 | 77.16 | - |
| | | $\sigma$ (%) | 1.67 | 1.42 | 2.34 | 5.79 | - |
| | | $C_v$ (%) | 10.53 | 5.65 | 5.65 | 7.50 | - |
| | 7-day UCS | $\bar{x}$ (MPa) | 0.00 | 0.00 | 0.43 | 1.22 | 1.89 |
| | | $\sigma$ (MPa) | 0.00 | 0.00 | 0.01 | 0.01 | 0.01 |
| | | $C_v$ (%) | 0.00 | 1.00 | 2.00 | 3.00 | 4.00 |
| | 28-day UCS | $\bar{x}$ (MPa) | 0.00 | 0.77 | 1.01 | 1.50 | 2.97 |
| | | $\sigma$ (MPa) | 0.00 | 0.02 | 0.01 | 0.01 | 0.02 |
| | | $C_v$ (%) | 0.00 | 1.00 | 2.00 | 3.00 | 4.00 |
| | 7-day ITS | $\bar{x}$ (MPa) | 0.00 | 0.00 | 0.00 | 0.18 | 0.37 |
| | | $\sigma$ (MPa) | 0.00 | 0.00 | 0.00 | 0.01 | 0.01 |
| | | $C_v$ (%) | 0.00 | 1.00 | 2.00 | 3.00 | 4.00 |
| | 28-day ITS | $\bar{x}$ (MPa) | 0.00 | 0.00 | 0.19 | 0.27 | 0.52 |
| | | $\sigma$ (MPa) | 0.00 | 0.00 | 0.02 | 0.01 | 0.02 |
| | | $C_v$ (%) | 0.00 | 1.00 | 2.00 | 3.00 | 4.00 |

**Table 4.** *Cont.*

| Tested Property | | | Hydraulic Binder Content (%) | | | | |
|---|---|---|---|---|---|---|---|
| | | | 0 | 3 | 5 | 7 | 9 |
| III batch of specimens (with S10–S15) | $W_L$ | $\overline{x}$ (%) | 42.55 | 43.20 | 43.37 | 43.23 | 43.34 |
| | | $\sigma$ (%) | 1.65 | 1.68 | 1.56 | 1.52 | 1.53 |
| | | $C_v$ (%) | 3.89 | 3.89 | 3.59 | 3.52 | 3.52 |
| | $W_P$ | $\overline{x}$ (%) | 29.96 | 30.13 | 30.16 | 30.09 | 30.18 |
| | | $\sigma$ (%) | 0.95 | 0.94 | 0.99 | 0.97 | 0.98 |
| | | $C_v$ (%) | 3.17 | 3.13 | 3.29 | 3.22 | 3.26 |
| | $I_P$ | $\overline{x}$ (%) | 12.59 | 13.07 | 13.21 | 13.14 | 13.16 |
| | | $\sigma$ (%) | 1.79 | 1.78 | 1.63 | 1.56 | 1.57 |
| | | $C_v$ (%) | 14.20 | 13.58 | 12.31 | 11.87 | 11.95 |
| | $W_{opt}$ | $\overline{x}$ (%) | 19.88 | 20.50 | 20.53 | 20.69 | 20.82 |
| | | $\sigma$ (%) | 0.80 | 0.73 | 0.90 | 0.88 | 0.89 |
| | | $C_v$ (%) | 4.05 | 3.57 | 4.38 | 4.27 | 4.26 |
| | $\gamma_{d.max}$ | $\overline{x}$ (Mg/m$^3$) | 1.61 | 1.67 | 1.70 | 1.72 | 1.75 |
| | | $\sigma$ (Mg/m$^3$) | 0.03 | 0.03 | 0.03 | 0.03 | 0.05 |
| | | $C_v$ (%) | 1.57 | 1.97 | 1.84 | 1.99 | 2.76 |
| | CBR | $\overline{x}$ (%) | 5.42 | 23.74 | 66.16 | 120.30 | - |
| | | $\sigma$ (%) | 1.25 | 3.92 | 3.37 | 3.72 | - |
| | | $C_v$ (%) | 23.13 | 16.53 | 5.09 | 3.09 | - |
| | 7-day UCS | $\overline{x}$ (MPa) | 0.00 | 0.00 | 0.42 | 1.17 | 1.87 |
| | | $\sigma$ (MPa) | 0.00 | 0.00 | 0.01 | 0.02 | 0.02 |
| | | $C_v$ (%) | 0.00 | 0.00 | 2.81 | 1.35 | 0.81 |
| | 28-day UCS | $\overline{x}$ (MPa) | 0.00 | 0.72 | 0.95 | 1.47 | 2.93 |
| | | $\sigma$ (MPa) | 0.00 | 0.01 | 0.01 | 0.01 | 0.01 |
| | | $C_v$ (%) | 0.00 | 2.05 | 1.20 | 0.78 | 0.39 |
| | 7-day ITS | $\overline{x}$ (MPa) | 0.00 | 0.00 | 0.00 | 0.15 | 0.33 |
| | | $\sigma$ (MPa) | 0.00 | 0.00 | 0.00 | 0.02 | 0.01 |
| | | $C_v$ (%) | 0.00 | 0.00 | 0.00 | 12.65 | 3.98 |
| | 28-day ITS | $\overline{x}$ (MPa) | 0.00 | 0.00 | 0.16 | 0.24 | 0.45 |
| | | $\sigma$ (MPa) | 0.00 | 0.00 | 0.01 | 0.01 | 0.01 |
| | | $C_v$ (%) | 0.00 | 0.00 | 5.30 | 4.83 | 2.88 |

The arithmetic mean of the results from individual properties of each batch of specimens was calculated using Equation (1):

$$\overline{x} = \frac{\sum_{i=1}^{n} x_i}{n} \tag{1}$$

where $\overline{x}$ is the arithmetic mean of "$n$" test results, $x_i$ is the value of each individual result from a total of "$n$" results, and $n$ is the number of specimens.

In addition to the arithmetic mean, the value of the standard deviation was determined according to Equation (2):

$$\sigma = \sqrt{\frac{\sum_{i=1}^{n}(x_i - \overline{x})^2}{n}} \tag{2}$$

where $\sigma$ is the standard deviation, $\overline{x}$ is the arithmetic mean of "$n$" test results, $x_i$ is the value of each individual result from a total of "$n$" results, and $n$ is the number of specimens. The standard deviation represents deviations of values from the examined mean value.

In order to obtain the overall statistics for each tested property per batch, the coefficient of variation was additionally calculated using Equation (3):

$$C_v = \frac{\sigma}{\overline{x}} \tag{3}$$

where $C_v$ is the coefficient of variation of results for each tested property per batch, $\sigma$ is the standard deviation, and $\overline{x}$ is the arithmetic mean of "$n$" test results. The coefficient of variation serves to determine the deviation from the arithmetic mean of the set, as well as the display of variability, i.e., whether it belongs to a homogeneous set ($C_v < 30\%$) or a heterogeneous set ($C_v > 30\%$). When samples are homogeneous, it is understood that the

samples are uniform, and a set of samples that are heterogeneous or have a value greater than 30% is non-uniform.

After the analysis, it is noticeable that in the batches I, II, and III, the consistency values ($W_L$, $W_P$, and $I_P$) have small deviations in terms of the mean value. By examining the coefficient of variation, the specimens have values that are less than 30% and represent uniform or homogeneous specimens. Regarding the coefficient of standard deviation, large differences in relation to the mean value occurred in the case of specimens from batch I (with S1–S5), namely for the liquid limit $W_L$ and the plasticity index $W_P$. For the plasticity index, there are large deviations in terms of standard deviation that occur in the case of specimens from batch III (with S11–S15). The smallest deviation related to the plasticity index occurs in batch II (S6–S10), while the minimum values of $W_L$ and $W_P$ occur in batch I (S1–S5). By analyzing the values of the optimal water content and maximum compaction, it was observed that with an increase in the amount of binder, there is an increase in the mean values of the maximum density, as well as the optimal amount of water. By reviewing the values of the standard deviation, it was observed that the specimens of batch III have the smallest deviations from the mean value at the maximum density, as well as the optimal amount of water. The maximum deviation in the coefficient of standard deviation was achieved in the specimens of batch II for the maximum density, and in the specimens of batch I for the maximum deviation in the optimal amount of water. Through statistical analysis, it was determined that the carrying capacity of the subgrade CBR is higher with increasing amounts of binder. From Table 4, it was observed that for the specimens with 9% binder, the value of the carrying capacity of the subgrade could not be recorded. Minimum deviations in CBR in terms of standard deviation were achieved with specimens of batch I, then batch II, and maximum deviations were registered with batch III. Statistical analysis has proven that, based on CBR values, all specimens are homogeneous. For specimens of batch I with 0% binder, the deviation is approximately 30%. The statistics of the UCS results after 7 and 28 days show that with an increase in the amount of binder, there is an increase in the unconfined compressive strength. After reviewing all the analyzed results, it was concluded that the specimens are homogeneous and that all UCS values are less than 30%. Observed through the standard deviation parameter, the specimens deviate from the mean value in the range of 0–0.02%, which does not represent a large deviation and implies that the specimens are homogeneous. It is known that the coefficient of variation directly depends on the mean value and the standard deviation. The ITS has a similar statistical analysis as the UCS test, i.e., with the increase in the amount of binder, the value of the ITS increases. At the initial values of 3% and 5%, no value could be measured because it was too small. Based on the statistics, it was found that the deviation from the mean value is quite small, i.e., it ranges from 0 to 0.02%, on the basis of which it can be concluded that the specimens are uniform.

After analyzing the results shown in Table 4, it was noticed that the standard deviation met all the requirements. The values of the variation coefficient were less than 30% (i.e., the homogeneous set). The only extreme value occurred for the CBR in the case of the second batch of specimens (with S1–S5) where the variation coefficient value was close to the limit of 30%.

Based on the presented results, it can be concluded that the mixtures that contain sandy components (specimens from the second batch) had the highest values of UCS, i.e., the values were higher by 2 to 5% compared to the UCS values of other batches.

*3.1. Consistency Limits*

Determination of the Atterberg limits was performed on all specimens, as well as the determination of the plasticity index, except in the case of mixtures that contained soil S6 and soil S7, i.e., they were characterized as the non-cohesive soils according to the results shown in Table 1. The tests were performed in accordance with standard SRPS U.B1.020 [38]. A test was conducted on different soil–binder mixtures, in order to see the trend of the limits as well as the plasticity index with and without binder material. The

results of the liquid limit test are shown in Figure 6a. It is noticeable that the values of the liquid limit had small increases as the amount of binder increased.

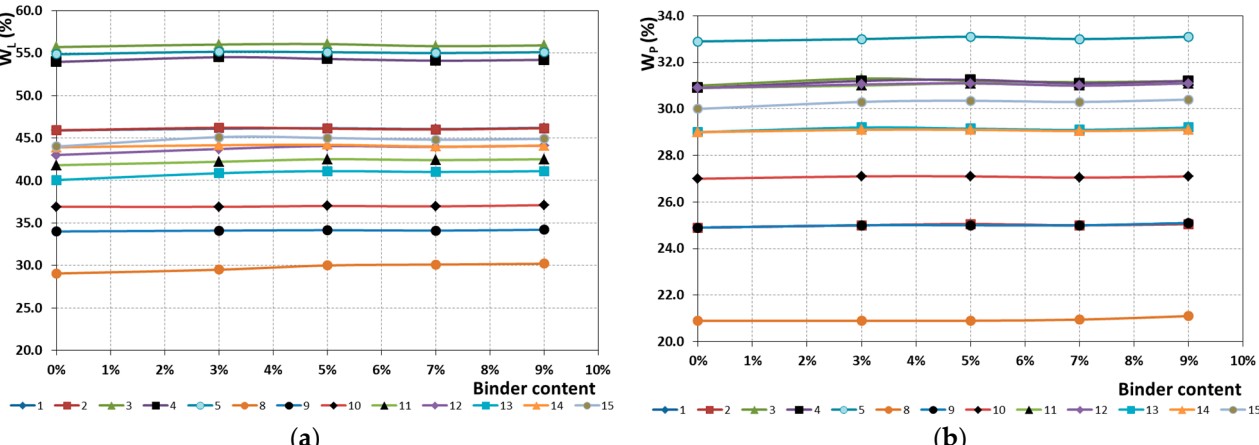

**Figure 6.** Variation in (**a**) the liquid limit and the binder content, and (**b**) the plasticity limit and the binder content.

The plasticity limit of the tested materials is shown in Figure 6b. Primarily, it is observed that the plasticity of the material was not recorded for some mixtures (containing S6 and S7), and it was also observed that increasing the amount of binder led to an increase in the plasticity limit in other mixtures. The minimum values were achieved for mixtures containing S8, S9, and S10 soils, while the maximum value was recorded for mixtures containing dusty clays (i.e., the first batch of soil–binder mixtures).

The plasticity index of the examined mixtures is shown in Figure 7. It can be observed that increasing the binder content up to a certain limit led to a slight increase in the plasticity index values, and after that they were constant. The plasticity index was reduced in mixtures containing soils belonging to the loess material group (i.e., the third batch of soil–binder mixtures), while the highest plasticity was achieved in mixtures containing soils that belong to the dusty clay group (i.e., the first batch of soil–binder mixtures).

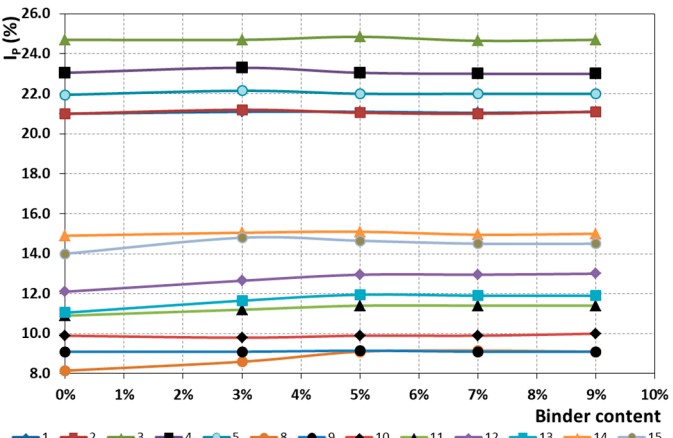

**Figure 7.** Variation in the plasticity index and the binder content.

*3.2. Compaction Characteristics (Proctor Compaction Test)*

The variation in $W_{opt}$ and $\gamma_{d,max}$ with hydraulic binder content is presented in Figure 8. Based on the presented results, it can be observed that there was an increase in $W_{opt}$ in the case of most mixtures containing up to 3% binder content, while at 5% binder content there was a decrease in the $W_{opt}$. Certain mixtures showed an extreme change in $W_{opt}$ with increasing binder content, like mixtures containing soils S8 and S9. In those mixtures, there

was a decrease in $W_{opt}$ at 3% binder content, and later, by increasing the percentage of the binder, there was only a certain increase in $W_{opt}$.

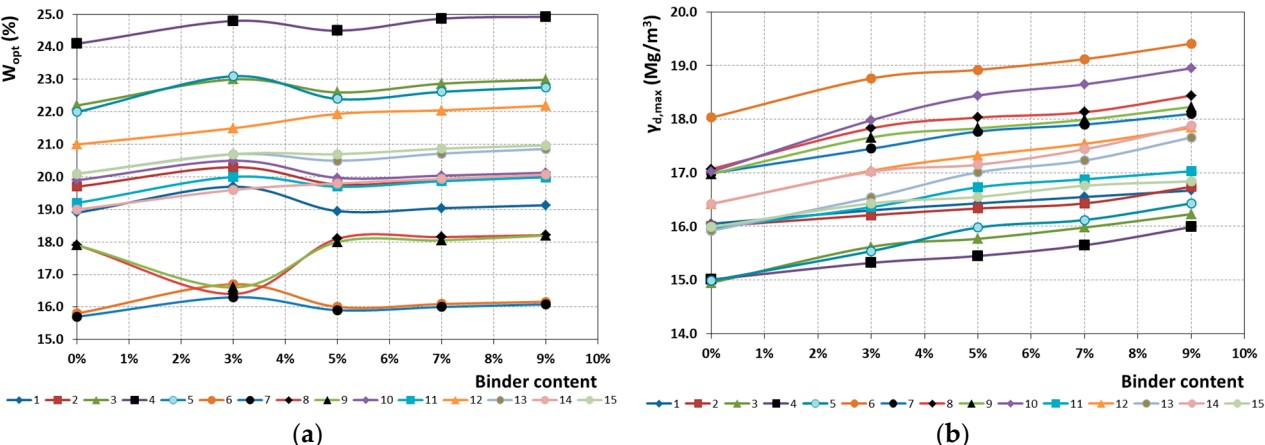

(**a**)        (**b**)

**Figure 8.** Variation in (**a**) $W_{opt}$ and the binder content, and (**b**) $\gamma_{d,max}$ and the binder content.

Figure 8b presents the variation in the $\gamma_{d,max}$ according to the amount of binder in the mixtures. By analyzing the obtained values, it can be concluded that increasing the binder amount led to an increase in $\gamma_{d,max}$. The maximum values were achieved for mixtures containing sandy materials or so-called non-cohesive materials, and the lowest values were recorded for mixtures with dusty clay materials.

*3.3. Bearing Capacity—California Bearing Ratio (CBR)*

CBR is a number that physically represents the indentation resistance of a standard piston in relation to standard values. It is one of the basic parameters of material quality when it comes to pavement construction design in the road sector. Figure 9 presents the variation in the CBR and the amount of binder in the mixtures. It can be observed that an increase in binder content brought an increase in the CBR.

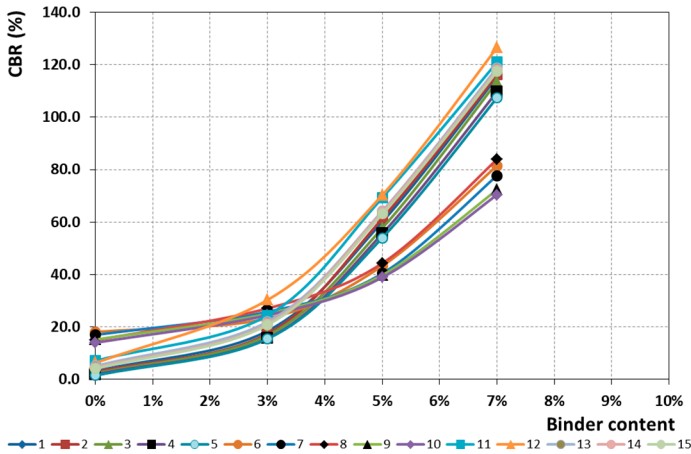

**Figure 9.** Variation in the CBR and the binder content.

Values of CBR with 9% binder contents were not recorded because the press could not apply enough pressure on any of the specimens. The above-mentioned indicates that the CBR values in this case were over 300% because that corresponds to the maximum limit of the press. The maximum values from the presented diagrams in Figure 9 belong to the mixtures with loess soils (i.e., the third batch of soil–binder mixtures), and the minimum values belong to the mixtures with sandy soils (i.e., the second batch of soil–binder mixtures).

Similar results were obtained by other authors. According to the research of Khemissa and Mahamedi [18], in terms of the CBR of an expansive overconsolidated clay treated with a mixture of various cement and lime contents and compacted under the optimum Proctor conditions, the mix treatment allowed researchers to increase the soaked and unsoaked CBR values, which led to an increase in the bearing pressure of the clay and a reduction in its expansibility. Okonkwo and Kennedy [17] also confirmed that both cement and lime were effective stabilizing agents that increased the CBR values of stabilized subgrade soil consisting of black cotton soil (an expansive clay soil type). Furthermore, a positive impact of powder lime (at the optimum level) in increasing the CBR value of treated kaolinite clay soil was confirmed by Tanzadeh et al. [13].

### 3.4. Unconfined Compressive Strength (UCS)

The UCS represents the most frequently investigated property that is used to evaluate the effectiveness of the stabilization with cement and/or other additives. In this study, the effect of different percentages of the applied hydraulic binder (based on cement and lime) on the mechanical behavior of the treated soils was evaluated through the UCS test. Figure 10a presents the UCS results after 7 days of specimen curing, while UCS results after 28 days of curing are presented in Figure 10b.

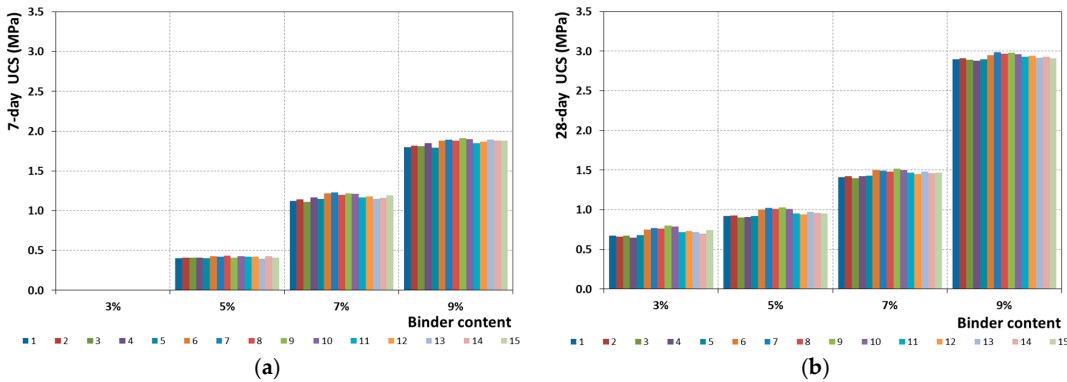

**Figure 10.** UCS results (**a**) after 7 days of curing, and (**b**) after 28 days of curing.

The 7-day UCS values of the specimens containing 3% binder could not be determined, while with the participation of binders from 5 to 9%, a polynomial increase with very strong correlation ($R^2 = 1$) in UCS was observed regardless of the soil type; see Figure 11a. After 28 days, it was possible to determine the UCS for the mixtures with 3% binder. Furthermore, with binder contents from 3 to 9%, a polynomial increase with strong corellation ($R^2 \sim 1$) in the 28-day UCS was observed, again regardless of the soil type; see Figure 11b.

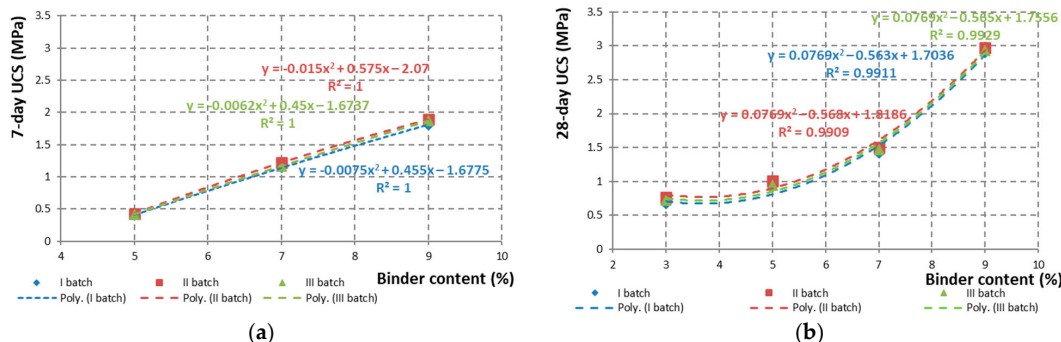

**Figure 11.** Correlation between arithmetic mean of UCS results for each batch of mixtures and the binder content (**a**) after 7 days of curing, and (**b**) after 28 days of curing.

It was observed that UCS values were higher with (i) an increase in hydraulic binder content based on a mix of cement and lime and (ii) an increase in the curing time. The formation of calcium-silicate-hydrate (C-S-H), a product of Portland cement hydration that contributes to the development of its strength, may further strengthen soil stabilized with calcium hydroxide produced as the by-product of cement hydration. The formation of C-S-H is characteristic of Portland cement, but not of lime. Additional C-S-H can form in both the Portland cement–soil and lime–soil systems due to the reaction between calcium hydroxide supplied by either cement or lime and the silica supplied by soil, and this process is known as a pozzolanic reaction. Calcium may also react with alumina and produce calcium-aluminate-hydrate (C-A-H). The formation of these additional C-S-H and C-A-H cementing materials may require the solubilization of silica and alumina from the soil components such as clay minerals, quartz, feldspars, and micas [6].

In the standard SRPS U.E9.024 [43], the requirements are given that must be satisfied for the mixture used in the production of road pavement construction bearing layers. Those requirements are shown in Table 5.

**Table 5.** UCS requirements for road pavement layers of different classes of roads in accordance with SRPS U.E9.024 [43].

| Layer | 7-Day UCS (MPa) | 28-Day UCS (MPa) |
|---|---|---|
| Base course and subbase course of the road pavement construction of highways and roads of classes I and II | 2–5.5 | 3–6.5 |
| Subbase course of the road pavement construction of class III and class IV roads | 1.5–4.5 | 2.5–6 |

It was observed that soil–binder mixtures made with 9% binder fulfilled the requirements in terms of 7-day and 28-day UCS, and therefore they could be used as a subbase course of the road pavement construction of class III and class IV roads.

The other authors have also confirmed the significant impact of increasing UCS values of the stabilized soil from the addition of binders up to a certain level. Okonkwo and Kennedy [17] confirmed that cement and lime were effective binders that increased the UCS values of subgrade expansive clay soil. Lebo et al. [7] reported that the utilization of cement, fly ash, and slag can improve the UCS of clay, depending on the amount of binder and the curing time of the mixture. A significant positive impact of lime in increasing the UCS value of treated kaolinite clay soil at the optimum level of lime powder was confirmed by Tanzadeh et al. [13]. Ghobadi et al. [15] reported that the optimum lime content and proper curing time for lime-treated clay soils were at least 7% and 30 days, respectively.

### 3.5. Indirect Tensile Strength (ITS)

Most granular materials cannot be subjected directly to tensile forces. For this purpose, and for the purpose of finding out about the possible acceptance of tensile forces, a standard-defined method of ITS was developed. From Figure 12, it can be observed that after 7 days of curing, the ITS could not be determined for mixtures with 3 and 5% binder, and after 28 days, the ITS value could be determined for the mixture with 3% binder. Furthermore, the ITS values were higher with increasing binder content and curing time of the mixture, and the highest ITS values were obtained for mixtures from the second batch, which contained sandy components.

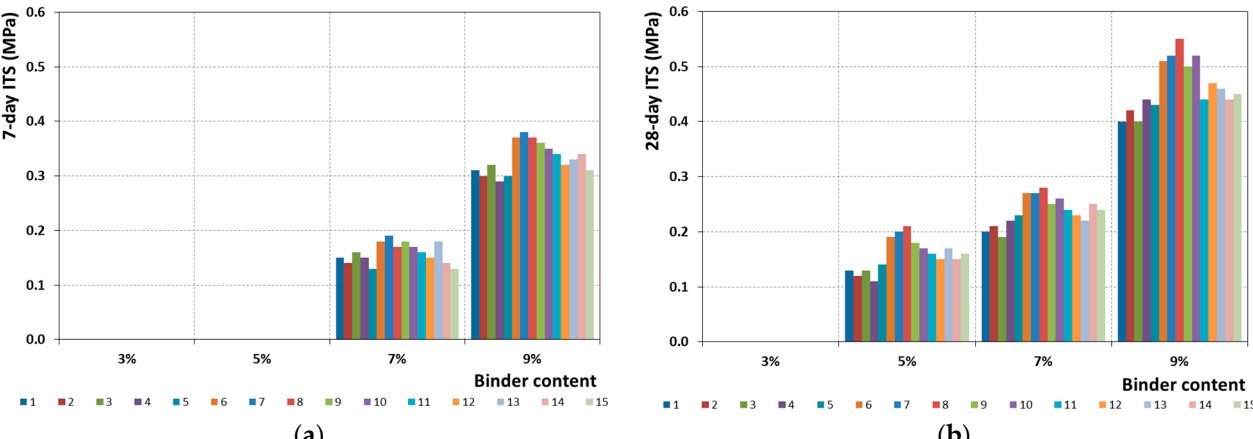

**Figure 12.** ITS results (**a**) after 7 days of curing, and (**b**) after 28 days of curing.

Figure 13 shows the correlation between the arithmetic mean of ITS results for each batch of mixtures and the binder content after 28 days of curing. With binder contents from 5 to 9%, a non-linear relationship in the form of a polynomial and with very strong correlation ($R^2 = 1$) in the 28-day ITS values was observed. The mixtures that contain sandy components (the second batch) had the highest ITS, followed by the mixtures with loess soils (the third batch) and the mixtures containing soils belonging to the dusty clay group (the first batch), respectively, regardless of the amount of the binder content.

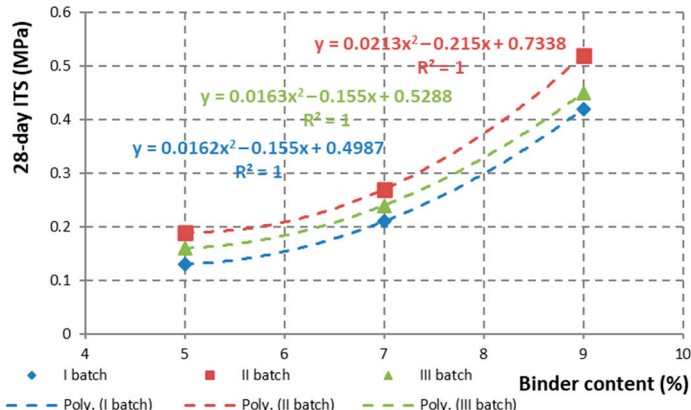

**Figure 13.** Correlation between arithmetic mean of ITS results for each batch of mixtures and the binder content after 28 days of curing.

*3.6. Comparison of Obtained Results with Other Literature Data*

After the review of the literature in the introduction and the presented research results in the paper, a comparative analysis of the obtained results with other literature data was carried out. The results of the comparative analysis are given in Table 6, which shows what has been done by testing in the literature and what results were obtained. By reviewing the papers, it can be seen that in [7], an analysis of soil stabilization was carried out, where UCS values were obtained after 7 and 28 days. No material plasticity data were available, but it is notable that the UCS results after 28 days are approximately similar to the results obtained in this paper. In [13], soil consistency tests were performed, where the plasticity values were significantly increased, and in addition, the results of the tests that were performed were reduced because pure lime was used for the binder in a percentage of 2 to 16%. The used soil material for the test was clay soil with medium to high plasticity. The authors of [14] analyzed sandy clay soil, mixed with lime binder in percentages of 3 to 9%, where the UCS results were lower, while the ITS values obtained after 28 days were approximately the same as the results in this paper. Some of the authors also used bio ash as a substitute

for cement [22], on the basis of which they achieved a significant result in the value of the placenta carrying capacity (CBR), up to 33.5%, and the values of UCS after 28 days are much lower. In [17], it was observed that the CBR values were significantly increased compared to other analyzed works, but the UCS values after 7 and 28 days were much less because they used pure cement and pure lime as binder. The authors of [19] also obtained smaller values of UCS after 7 and 28 days in their work, but in addition, they failed to improve the load-bearing properties of the placenta through CBR.

**Table 6.** Comparative analysis of obtained results with other literature data.

| Papers | $W_L$ | $W_P$ | $I_P$ | $W_{opr}$ | $\gamma_{d.max}$ | CBR | 7-Day UCS | 28-Day UCS | 7-Day Its | 28-Day Its |
|---|---|---|---|---|---|---|---|---|---|---|
| [7] | - | - | - | - | - | - | 0.12–1.52 | 0.12–3.00 | - | - |
| [12] | 67.8–77.2 | 40.8–51.1 | 18.2–33.3 | 25.9–33.7 | 1.17–1.68 | 2.3–14.3 | 0.48–0.75 | 0.52–0.84 | - | - |
| [13] | 53.1 | 31.8 | 21.3 | 30–32.5 | 1.35 | - | - | 0.32–1.207 | - | 0.06–0.245 |
| [16] | 16.65–37.9 | 13.94–20.6 | 2.71–15.4 | 12.6–15.7 | 1.53–1.99 | 6.3–82.88 | 0.23–0.30 | 0.28–0.36 | - | - |
| [18] | 18.98–23.9 | 28.67–34.78 | 9.13–10.9 | 6.31–8.94 | 2.13–2.41 | 2.9–10.12 | 0.11–0.15 | 0.12–0.17 | - | - |
| [21] | | | | 1.70–1.78 | 8.0–21.0 | 2.5–33.5 | - | 0.30–0.90 | - | - |
| Authors paper | 29.05–55.7 | 20.9–32.9 | 8.15–24.7 | 15.7–24.1 | 1.49–1.80 | 1.5–122 | 0.40–1.95 | 0.6–3.0 | 0.13–0.37 | 0.12–0.57 |

Based on reviewing the entire paper and presenting a comparative analysis of the results with other literature data, it can be concluded that in this paper, cement- and lime-based binders were used and significant improvements were shown in the carrying capacity of the subgrade over the CBR value. In addition, the results obtained with UCS after 7 and 28 days show that these binders have a very good effect on soil stabilization. It should be taken into account with these binders that the results are significantly improved with clay materials of medium and high plasticity, as well as with dusty clay materials where we have a much-reduced share of sand. In the case of sandy and gravelly materials, stabilization does not have a great improvement because these materials have their own natural carrying capacity.

## 4. Conclusions

Based on the presented experimental results of determining Atterberg limits with the plasticity index, $\gamma_{d,max}$, $W_{opt}$, CBR, UCS, and ITS of different soil types taken from 15 different locations in Vojvodina (Serbia) that were stabilized with 3, 5, 7, and 9% of hydraulic binder based on a mix of cement and lime, the following conclusions can be drawn:

- The soil samples were classified into three groups (i.e., batches) according to the granulometric composition: (i) the dusty clay materials (cohesive soils), (ii) the sandy materials with much less clay particles (non-cohesive soils), and (iii) the clays of low plasticity or the so-called loess material (cohesive soils). Soil samples were sampled and examined in order to cover the entire area of Vojvodina and to obtain basic data on the possible stabilization of the soil subgrade, which is of great importance for planning the road network.
- Increasing the amount of binder did not significantly affect the consistency of the stabilized soils, but the results had a gradual growth trend.
- The Proctor's compaction test results showed that increasing the amount of binder led to an increase in $\gamma_{d,max}$, and in most of cases an increase in $W_{opt}$, except for soil–binder specimens with a sandy component, where certain oscillations occurred in the results.
- The bearing capacity of the soil–binder specimens was evaluated by CBR, where specimens were tested only for pressing pistons up to 2.54 mm. The maximum CBR values with 5 and 7% had the mixtures with loess soils, and the minimum values belonged to the mixtures with sandy soils. With an increase in the binder content there was an increase in the bearing capacity of the stabilized soils. In the case of the specimens that had a binder addition of 9%, the CBR could not be determined, because the press could not apply enough pressure on those specimens, which confirmed that these materials had CBR > 300%, i.e., corresponding to the maximum limit of the press.

- USC values depended primarily on the curing time of the soil–binder specimens, then on the percentage of added hydraulic binder, and least on the type of used soil. The 7-day UCS values of the specimens containing 3% binder could not be determined, while with binder contents from 5 to 9%, a polynomial increase with very strong correlation ($R^2 = 1$) in UCS was observed regardless of the applied soil type. With binder contents from 3 to 9%, a similar conclusion was observed in the 28-day UCS, again regardless of the applied soil type.
- The amount of binder had a dominant effect on the ITS of soil–binder specimens, followed by the length of specimen curing time, and the type of used soil, respectively. The ITS could not be determined for mixtures with 3 and 5% binder after 7 days of curing or for the mixture with 3% binder after 28 days of curing. Generally, ITS values increased with increasing binder content and increasing curing time of the mixture, and the highest ITS values were obtained for soil–binder mixtures that contained sandy components.
- Based on the presented results, stabilization can be recommended for different types of soils from a total of 15 different locations in Vojvodina with a hydraulic binder added in the amount of 3–9%, as well as their further usage as a subgrade. It is necessary to conduct additional soil stabilization tests with different amounts of hydraulic binder in order to obtain a more complete picture of the relevant data for the area where the construction of the specific road is planned and to determine the optimum binder content in that particular case.

**Author Contributions:** Conceptualization, T.M. and M.Š.; methodology, F.K. and N.R.; validation, A.S.-Ć., V.B. and F.K.; formal analysis, F.K.; investigation, F.K., A.S.-Ć., V.B., T.M. and M.Š.; resources, F.K. and M.Š.; data curation, F.K. and M.Š.; writing—original draft preparation, T.M., M.Š., A.S.-Ć. and V.B.; writing—review and editing, T.M., V.B., N.R. and A.S.-Ć.; visualization, M.Š.; supervision, A.S.-Ć. and V.B. All authors have read and agreed to the published version of the manuscript.

**Funding:** This research received no external funding.

**Data Availability Statement:** The data presented in this study are available in the paper.

**Acknowledgments:** This paper has been supported by the Ministry of Science, Technological Development and Innovation through project no. 451-03-47/2023-01/200156 "Innovative scientific and artistic research from the FTS domain".

**Conflicts of Interest:** The authors declare no conflict of interest.

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
