# Peer review of "Stabilization of Different Soil Types Using a Hydraulic Binder"

_buildings, doi:10.3390/buildings13082040_

Round 1

Reviewer 1 Report

The paper presents a valuable research on 15 different soils' stabilization, realized through cement and lime as hydraulic binder, with 5 different amounts of the binder (0%, 3%, 5%, 7%, and 9%), for a total amount of 75 mixtures.

Too low references for such a well-known issue. You have to read and improve the state of art with several other papers, such as:

·         Ahmad, A., Sutanto, M. H., Ahmad, N. R. B., Bujang, M., & Mohamad, M. E. (2021). The implementation of industrial byproduct in malaysian peat improvement: A sustainable soil stabilization approach. Materials14(23), 7315.

·         Caselles, L. D., Hot, J., Roosz, C., & Cyr, M. (2020). Stabilization of soils containing sulfates by using alternative hydraulic binders. Applied Geochemistry113, 104494.

·         Makusa, G. P. (2013). Soil stabilization methods and materials in engineering practice: State of the art review.

Lines 39-44 are too much repetitive. Efforts have been made to introduce just lime and cement, but several hydraulic and non-hydraulic materials exist to stabilize natural soils. There are several commonly used binders, such as: cement, lime, fly ash, blast furnace slag, or pozzolans. It could be interesting to present an overview on all of them, and then to focus just on the hydraulic ones and say why. It is also interesting to know the main differences between them, especially between lime and cement. 

Table 1 for the soil consistency, and classifications (both for the AASHTO and USCS) it is necessary to introduce the abbreviation meaning into the text (i.e. WL, WP, IP, A-7-6, or CL and so on).

The Figures description in the text is absolutely not clear. The “Figure” word is introduced most of the times at the end of statements, without any sense.

The statistical analysis of the obtained results is weak and not deeply discussed. It is not enough just trying to explain the main characteristics of the 3 batches just with the arithmetic mean, the standard deviation, and the coefficient of variation of results. What is the homogeneous set and a coefficient variation value less than 30%? You have to explain it in a better way.

In general, have you got just 1 sample for a given soil and a given % of binder? I think it is too unrepresentative to obtain statistical valuable results.

English must be improved.

There are a lot of inaccurancies related to the punctuation, the subjects of a lot of sentences (like lines 147 and 149), the articles, and so on. A grammar extensive review has to be done.

Author Response

The authors appreciate and show gratitude for all the comments given by Reviewer. All added or changed text in the manuscript is marked with blue.

Reviewer 2 Report

The authors have presented an interesting paper for the construction sector, with up-to-date information and well-defined tests. It is understood that the work is suitable for the Buildings journal, and that only some minor issues need to be resolved.

Reduce the number of keywords included.

The introduction presents a careful and detailed review of the current state of knowledge, providing relevant references and discussing them in depth. 

Line 55 use the same number of significant figures.

Line 98 and others, remove the + sign.

Lines 152 to 159, briefly explain the tests carried out and include the applicable regulations, equipment and other relevant information. Idem, lines 174 to 179. If they are to be explained later, delete the dotted numbering, it is not necessary.

Equation (1) is well known and can be deleted. Ditto with equations (2) and (3).

The presentation of the results graphs could be improved. Additionally, these could be disucssed on the basis of previous studies.

Include possible limitations of the research at the end of the manuscript.

Author Response

The authors appreciate and show gratitude for all the comments given by Reviewer. All added or changed text in the manuscript is marked with yellow.

Round 2

Reviewer 1 Report

The authors have clearly modified the paper as suggested. For me, it is now suitable for publication.

Thanks a lot.